# Exploring the Epidemiological Surveillance of Hepatitis A in South Africa: A 2023 Perspective

**DOI:** 10.3390/v16060894

**Published:** 2024-05-31

**Authors:** Keveshan Bhagwandin, Jayendrie Thaver-Kleitman, Kathleen Subramoney, Morubula Jack Manamela, Nishi Prabdial-Sing

**Affiliations:** 1Division of the National Health Laboratory Services, National Institute for Communicable Diseases, Johannesburg 2131, South Africa; keveshanb@nicd.ac.za (K.B.); jayendriet@nicd.ac.za (J.T.-K.); kathleens@nicd.ac.za (K.S.); jackm@nicd.ac.za (M.J.M.); 2School of Pathology, Faculty of Health Sciences, University of the Witwatersrand, Johannesburg 2000, South Africa

**Keywords:** hepatitis A, endemicity, surveillance, NMCSS, SDW

## Abstract

Hepatitis A (HAV) presents a significant global health concern with diverse clinical manifestations primarily transmitted through fecal–oral routes, emphasizing the critical role of sanitation and water cleanliness in transmission dynamics. Age-related variations, notably asymptomatic presentation in children, add complexity. The World Health Organization’s (WHO) endemicity classification aids in understanding prevalence and control strategies. This study examines 2023 South African epidemiological data on HAV cases, evaluating age distribution, incidence rates, and provincial disparities. Data from the national surveillance system and weather services were analyzed. Findings reveal distinct age-related trends, with the highest seroprevalence observed in the 5–9 age group with the most burdened areas located in the Western Cape, KwaZulu-Natal, and Gauteng provinces. Furthermore, seasonal rainfall variations correlate with increased incidence in Western Cape and KZN. The amalgamation of results suggest a potential epidemiological shift, emphasizing the need for updated immunization strategies. Noteworthy patterns, like the rise in 5–9-year-olds, may be influenced by factors such as school clustering and migration. Provincial disparities and the impact of climatic events underscore the necessity for dynamic vaccination strategies and surveillance network enhancements. This study highlights the urgency for improved understanding and response to HAV in South Africa.

## 1. Introduction

Hepatitis A is the most common cause of viral hepatitis globally, with clinical presentations varying from asymptomatic to mild and in some cases severe acute liver disease [1]. Hepatitis A virus (HAV) is classified as an enteric virus where transmission occurs through fecal–oral routes amongst infected individuals, contaminated food, and water [1,2]. Symptoms most common in adults include fever, malaise, dark urine, jaundice, nausea, and abdominal pain that may develop over a period of 1–2 months [1,3]. Around 70% of children below the ages of 6 years present as asymptomatic or may present with symptoms other than jaundice [4]. Symptoms are not just more common in adults; they also tend to be more severe with presentation of jaundice being observed [4]. 

The World Health Organization (WHO) through its position paper has illustrated various levels of endemicity by immune responses to HAV [1]. The WHO has outlined a highly endemic area as one that has a seroprevalence (total Ig or IgG count) > 90% by age 10, whereas an area with intermediate endemicity displays seroprevalence < 90% by 10 years old but by age 15 reaches > 50%. A low area of endemicity exhibits a seroprevalence < 50% by 15 years of age but increases to > by age 30, and a very low endemic area has a seroprevalence of <50% by 30 years old [1].

Previously, South Africa was highly endemic for hepatitis A; however, substantial improvements in water sanitation and sources could have potentially affected the epidemiology of HAV in South Africa [5]. An investigation conducted by Enoch and colleagues presented a surveillance study between 2009 and 2014 that identified a prominence of HAV seroprevalence (IgG and IgM) by age six [5]. A cross-sectional study showed that in the Western Cape Province (WCP), there was a significant drop in seroprevalence to approximately 44% in individuals aged 1–7 years old [5]. The latter study showed that seroprevalence increased as age increased, where 1–2-year-olds had a seroprevalence of 22% (45/203) while children aged 5 to 7 years had a seroprevalence of 62% (136/218) [5]. Contributing factors to the latter study were the improved water processing and sanitation in the WCP and socio-economic development in conjunction with amended hygiene practices in resource-poor regions [5]. To further substantiate the endemicity of hepatitis A in South Africa, a study undertaken from 2017 to 2020 uncovered high levels of seroprevalence amongst children up to the ages of 9 years old, with the Western Cape having the highest incidence compared to other provinces in South Africa [6]. 

Hepatitis A exists as a Notifiable Medical Condition (NMC) category 2 in South Africa and requires notification to the Department of Health (DoH) by clinicians and testing laboratories approximately 7 days after clinical or laboratory diagnosis. Surveillance datasets provide crucial information on age trends in disease incidence to monitor incidence as well as rapidly detect outbreaks for prompt response to disrupt the chain of transmission. Discussion centering around the implementation of an HAV vaccine in the Extended Immunization Program (EPI) has been mentioned in many papers prior [7,8]. This investigation describes the HAV epidemiological data derived from the Notifiable Medical Condition Surveillance Systems (NMCSS) and the National Health Laboratory Service (NHLS) Surveillance Data Warehouse (SDW) in 2023 across the nine provinces of South Africa.

## 2. Materials and Methods

Hepatitis A case information was investigated from two sources between 1 January to 31 December 2023. The SDW provided positives, negatives, and overall testing rates with respect to anti-HAV IgM results whereas the NMC data record positive anti-HAV IgM cases from private and public health care facilities and laboratories. Although different assays are used across NHLS laboratories, this did not impact our results as we mined data on positive IgM results as reported on the laboratory information system Additionally, the NMCSS provides epidemiological information including but not limited to clinical presentations, occupation, and a list of close contacts that the patient was in contact with. Data from the SDW and NMC were de-duplicated by patient ID numbers and folder numbers. Furthermore, weather reports and articles were used as supplemental information to determine if certain water-related events could potentially influence hepatitis A case fluctuations. Data were analyzed by age, gender, province, and rainfall patterns (Microsoft Excel 2022). The analysis of rainfall was conducted by looking at the average rainfall provided by the South African Weather Services (SAWS) in the targeted region per month which was plotted against cases that were confirmed IgM positives in that month. Additionally, a regression analysis was performed to identify if there was any association present.

## 3. Results

### 3.1. SDW Data for Anti-HAV IgM Testing in 2023

The overall number of cases tested in the country was 168,317, with 4607 positive cases and a positivity rate of 2.81% as seen in Table 1. WCP had the lowest testing rate but highest incidence (18/100,000), equating to a 9.27% laboratory positivity rate. The SDW data showed a lower incidence in the same period of 4457, partly due to clinical notifications not captured in the SDW data. Western Cape Province reported the highest incidence of cases (1457/4457; 32.70%), with a decrease in cases observed in January, April, and September, and a peak in cases observed in February–March (310 positive cases) and May–July (422 positive cases) (Figure 1). KwaZulu-Natal followed with 19.74% (880/4457) reported cases, with increases in incidence between January–February (122 positive cases), April–May (113 cases), June–August (189 cases), and September–October (173 positive cases). Similarly, Gauteng was the third significant contributor to cases in the country, accounting for an estimated 17.28% (770/4457) cases. Within Gauteng, cases oscillated between January–February (73 cases), April–May (102 positive cases), June–August (179 cases), and September–October (129 cases). Limpopo province reported 400 cases. The remaining cases are distributed among the remaining five provinces collectively, each reporting relatively lower numbers ranging from 127 cases (Northern Cape) to 245 cases (Eastern Cape) for the year.

Notably, there was a significant incidence in the younger population (Figure 2). Children aged 0–4 years accounted for 494/3581 (13.80%) cases, while those aged 5–9 contributed 975/3581 cases (27.23%), 10–14-year-olds comprised approximately 597/3581 (16.5%) cases, and 15–19-year-olds constituted around 323/3581 (9.01%) cases. Furthermore, adults aged 20–24 years contributed to 8.40% (301/3581) of cases.

### 3.2. NMCSS Pronvincial Data for Hepatitis A

Age was reported for 4457 cases with ages spanning from <1 to 75+ years old. The mean age was calculated to be 21 years with the median age group being the 20–24-year cohort. Furthermore, the inter quartile range (IQR) was from 5 to 23 years of age, with the highest proportion of cases clustering toward the 5–9-year age group (27.43%) followed by the 10–14-year-olds (16.83%). Significant patterns emerged, with children aged 0–4 showing the lowest prevalence (14%) and 35–75+-year-olds showing a consistent seropositivity rate of 90%.

With respect to gender distribution, similarities in numbers between genders can be observed across the different provinces with noticeable differences being observed in Gauteng, Kwa-Zulu Natal, and Limpopo provinces, where HAV has been detected more in males as illustrated in Figure 3. Statistical analysis between the male and female genders showed no significant difference with regard to HAV infection using a paired two-tailed *t*-test (*p* > 0.05, *p* = 0.18).

### 3.3. Linking Rainfall Patterns to Hepatitis A Cases 

Figure 4a–c describes the number of new Hepatitis A cases in a given month to the average rainfall in that same month for the most burned Hepatitis A provinces in South Africa. Additionally, a linear regression for each province can be observed below. In the Western Cape, there were three periods of markable rainfall increase (January–March, April–June, and August–September). A corresponding increase in HAV cases were displayed in 2/3 rainfall increase periods, which can be seen in January–February and April–March. Kwa-Zulu Natal had four periods of significant rainfall increases (January–February, April–May, September–October, and November–December) with analogous increases seen in all rainfall periods except December. Lastly, Gauteng had two notable increases in rainfall (January–February and September–December), with comparable increases in these intervals. A regression analysis and Pearson correlation coefficient was conducted to evaluate correlation between the two variables. Positive associations were observed in the Western Cape and KZN, with ρ values being 0.57 and 0.35, respectively. Conversely, negative association was observed in Gauteng with a ρ value of −0.10. 

## 4. Discussion

The study looked at hepatitis A incidence in the country in 2023. Cases of HAV have been predominately observed in the Western Cape (36.4%), KwaZulu-Natal (18.8%), and Gauteng (15.1%). This is further reinforced by the positivity rates. Of concern, the Western Cape, with the lowest testing rate, produced the highest positivity rate of 9.27%, followed by KZN (2.36%) and Gauteng (2.06%). Previous laboratory surveillance investigations reported similar findings on these provinces as major contributors to HAV seroprevalence in the country [7,8].

Climatic events have been correlated with increase in HAV cases, as seen in Spain in 2010–2014 [9]. Based on information supplied by the South African Weather Service (SAWS), elevated levels of rainfall throughout various parts of the country in 2023 were observed. With Gauteng experiencing heavy rains in its summer months, flash flooding is a common occurrence that usually arises when rainfall exceeds drainage capacity [10]. Interestingly, an associated increase in HAV positive cases was observed in KZN and Gauteng during the increase in rainfall. Coincidentally, a natural state of disaster was issued in relation to flash flooding and heavy rains in KZN and Gauteng between January and February [11]. Additionally, even in cases of minor elevations in rainfall between months (KZN: April–May and June–July; Gauteng: April–May) saw an increase in HAV cases in both provinces. In relation to the Western Cape, a minor increase in rainfall between January and February saw a major increase in cases with correlating elevation in both variables observed between April and May. Heavy rains were reported in the first quarter of 2023 [12], which possibly accounted for the resurgence in HAV cases within the province. However, after the initial rise in cases in the Western Cape, a continuous decline was noted throughout the course of the year. Intriguingly, an increase in HAV cases was further identified during seasonal rain caused by the El Nino effect throughout each of the provinces, with a plateau of cases seen in the Western Cape. Through the aforementioned data elaboration, it can be concluded that rainfall patterns and other climate change patterns need to be considered and monitored when addressing HAV and other enteric pathogen outbreaks, and our findings are further supported by Benson et al. [13]. However, flash flooding and rainfall patterns are not the sole contributor to fluctuations in HAV cases as there are instances where rainfall has increased without a direct rise in HAV cases. A statistical correlation and regression between rainfall and monthly HAV positive cases inferred a strong positive association in the Western Cape and a weak positive association in the KZN province. Gauteng showed a weak negative association between rainfall and HAV cases. Interestingly, a positive association was observed in coastal provinces, and these results imply that rainfall did to an extent have an influence on HAV cases; however, the weak negative association described in Gauteng alludes to the circulation and infection of HAV during an outbreak being influenced by multifactorial events, rather than just rainfall. It must be noted that a reduction of 150 cases between SDW (4607) and NMC (4457) can be accounted for by the fact that not all positive cases reported on SDW were reported on NMC.

In relation to endemicity, it can be observed that by the age of 14 years old, 58.40% of the sample population already exhibits anti-HAV IgM. Some of our data corroborate with Prabdial-Sing [9]. This can be observed by the fact that their study, in 2018, showed a high incidence of 21% in the 0–4-year age group, whereas our study showed only 14% in the same age group, in 2023. Incidence for age groups 5–9 and 10–14 were almost similar to that of the previous study [9], at 25% and 16%, respectively. Additionally, no significant discrepancies were observed between genders, with the slightly elevated numbers being apparent in males within Kwa-Zulu Natal, Gauteng, and Limpopo provinces. With more than half of our study population displaying exposure to the Hepatitis A virus by age 10–14 years, one can suggest that South Africa could have possibly undergone an epidemiological shift from a status of high to intermediate endemicity This is supported by consistent IgM results produced in Enoch’s study in 2014, highlighting the epidemiological transition that South Africa has potentially undergone as per the WHO criteria [5]. This study showed a similar trend occurring when anti-HAV IgM was analyzed in 2023. Older age groups of up to 35–39 years accounted for more than 90% of the study population with anti-HAV IgM seropositivity. A similar incidence of 89% was reported up to this age group in 2018 [9]. 

It is apparent that the 5–9 age group contributed to a significant portion (27.23%) of reported HAV cases in the country. HAV is most contagious 2 weeks before onset of symptoms; hence, with knowledge that this age group is currently schooling, it can be deduced that close contact with unknown infected individuals amalgamated with the absence of an HAV vaccine in the EPI can result in a cluster of outbreaks. The markable decline of IgM positives in the 1–4- and 10–14-year age groups, respectively, may be attributed to the fact that children in the age group 1–4 years are not at school yet nor in close contact with other children and the 10–14 age group have had mild infection previously and are now immune. Furthermore, potentially migration factors due to various socio-economic reasons could account for this specific data trend present. The possible influx of individuals from areas of high HAV incidence to relatively lower areas can possibly account for what was described in 2023; however, this would require further analysis to confirm. Data used in this paper highlighted several limitations that are present with the NMCSS. This was further substantiated in a publication where issues were included but not limited to underreporting, report completeness, and unequal coverage of NMC utilization across the country [13]. Potential improvements in the NMCSS can be through proper training of the system, enhancing data completeness, and focusing on user-related barriers to reporting and expansion of geographic coverage. 

## 5. Limitations 

The source of the high cases of HAV in the Western Cape, KZN, and Gauteng provinces was unknown and requires further investigation. Possibilities could include but were not just limited to contaminated food, water, heavy precipitation, and/or unmaintained wastewater plants. This data represented HAV diagnoses after clinical suspicion of disease; hence, it could be an under-representation as not all infected persons present with symptoms at healthcare facilities. Furthermore, this study only had access to IgM results, and IgG/total Ig results were not captured; hence, any inferences on geographic distribution using IgG could not be made. This is due to the fact that NMCSS is used for the communication of Category 1, 2, and 3 diseases. Therefore, all positive samples in our study were indicative of acute infections. Notes on clinical cases, such as symptoms, outcome of patients, and disease severity, etc., on the SDW and NMCSS were not reported on as the variables are often not captured at the clinical facility. 

## 6. Conclusions

This study has highlighted several plausible explanations for trends in incidence related to certain age cohorts, geography, and rainfall patterns. The incidence varied, with Western Cape, KZN, and Gauteng accounting for a significant portion of the cases. Ongoing monitoring from reliable data sources is pivotal for HAV surveillance in the country to reflect any further decline in immunity from natural infection at the young age groups, especially as we know that South Africa is in a possible epidemiological transition phase and may require the HAV vaccine/s in its EPI schedule; which may prove relevant in the future. The WHO position paper on hepatitis A supports the vaccination against HAV at the ages of 12 months and older on the foundations of (i) an increasing trend of acute disease over time in older children, adolescents, and adults, (ii) alterations in endemicity from high to intermediate, and (iii) considerations of cost-effectiveness [1]. Our results do support the need for implementation of an HAV vaccine in South Africa’s EPI for children aged 12 months and older. Therefore, effective protection against HAV can been seen as the culmination of environmental improvements and vaccination. Identified weaknesses in the surveillance network underline the urgency for further development and enhancement to better understand and respond to HAV. Furthermore, strengthening of the surveillance network around South Africa can be achieved through quarterly district reports, epidemiological mapping of genotypes and subtypes, as well as wastewater surveillance of hepatitis A in high burden areas and public health settings.

## Figures and Tables

**Figure 1 viruses-16-00894-f001:**
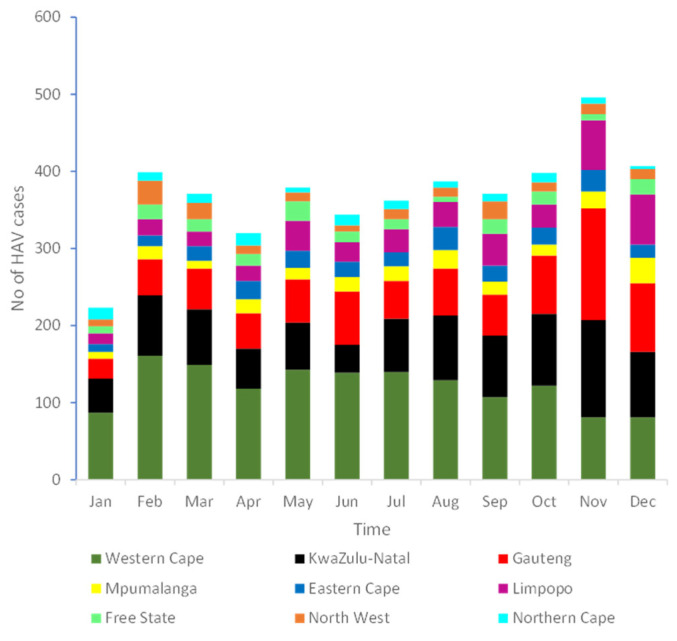
Number of anti-HAV IgM cases detected across South African provinces as supplied from the NMCSS.

**Figure 2 viruses-16-00894-f002:**
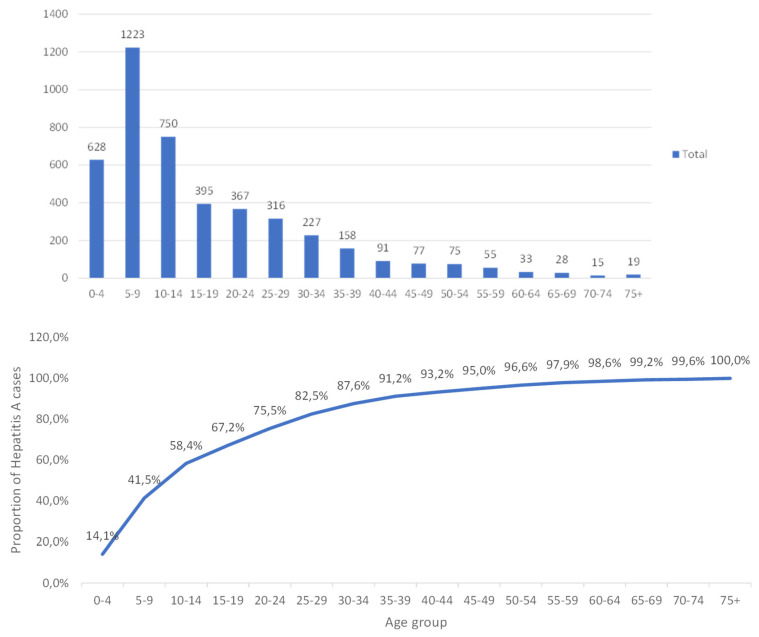
Age distribution and proportion of anti-HAV IgM in South Africa. Bar graphs represent each age group. The absolute number above the bars represent the total number of positive HAV cases in that age group. The proportion of positive cases are represented by the line graphs for each age group.

**Figure 3 viruses-16-00894-f003:**
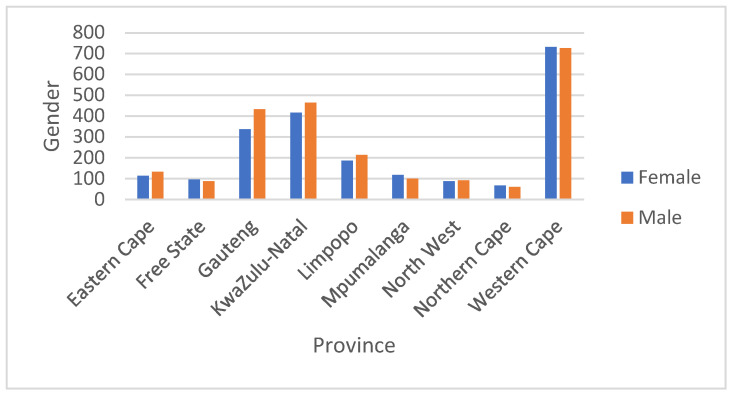
Distribution of anti-HAV IgM positive cases between genders.

**Figure 4 viruses-16-00894-f004:**
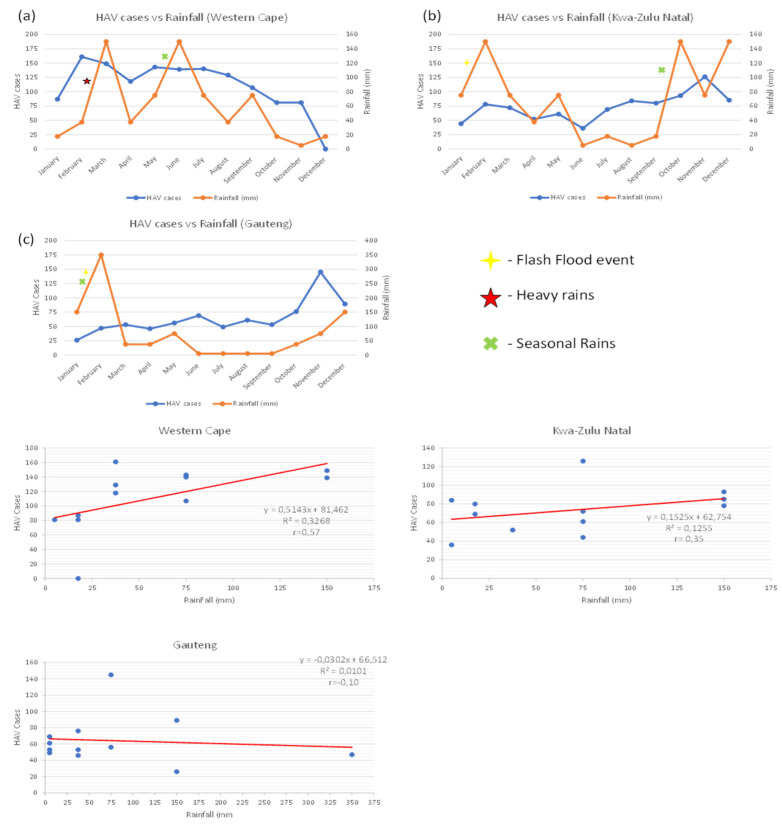
(**a**–**c**) Illustration and relation of monthly rainfall compared to HAV cases in the 3 most burdened provinces.

**Table 1 viruses-16-00894-t001:** Illustration of the testing rates across the different provinces and South Africa for 2023 as supplied from the SDW.

Province	Population	HAV-IgM Negative Cases	HAV-IgM Positive Cases	Total Tests	Lab Positive Rate (%)	Incidence Rate/100,000	Testing Rate/100,000
Eastern Cape	6,734,001	16,001	288	16,289	1.77	4	238
Free State	2,928,903	6430	150	6580	2.28	5	220
Gauteng	15,488,137	47,679	984	48,663	2.02	6	308
Kwa-Zulu Natal	11,531,628	39,251	927	40,178	2.31	8	340
Limpopo	5,852,553	13,463	426	13,889	3.07	7	230
Mpumalanga	4,679,786	12,245	251	12,496	2.01	5	262
North-west	4,108,816	9581	194	9775	1.98	5	233
Northern Cape	1,292,786	5278	109	5387	2.02	8	408
Western Cape	7,005,741	13,782	1278	15,060	8.47	18	197
South Africa	59,622,351	163,710	4607	168,317	2.71	8	275

## Data Availability

No new data were created.

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
