# Peer review of "Exploring the Epidemiological Surveillance of Hepatitis A in South Africa: A 2023 Perspective"

_viruses, 2024, doi:10.3390/v16060894_

Round 1

Reviewer 1 Report

Comments and Suggestions for Authors

Thank you for the opportunity to review this paper on the exploration of the epidemiological surveillance of hepatitis A in South Africa in 2023. Although the study reports several potentially interesting findings, I have a number of major concerns regarding the description of the data, the analysis and followed methodology, which influence the validity of the conclusions that may be drawn. More specifically, in the Methods (ln 72-73), the authors state that “The SDW provided positives, negatives and overall testing rates”. Positive for what marker? Total anti-HAV, IgG? And how were these tests done? Did all involved laboratories in all provinces use the same assay? Which one? Please specify and comment. This could be an important limitation of the study, which, in the very least, should be mentioned and discussed. Furthermore, in the next line the authors state “the NMC data records positive IgM cases from private and public health care facilities and laboratories.” Where are these results (IgM cases) shown? Are they included in the presented data and analysis? Are all presented results based on IgM tests, as mentioned much later, in the study limitations? If that is the case, that is, indeed, a major limitation of the study. Confidence intervals are not provided anywhere in the manuscript.

Minor, but still important points.

1. The disease (hepatitis A) should be clearly distinguished from the causative agent (hepatitis A virus, HAV). The two are used interchangeably in the text (e.g., ln 10, ln 29) and this is incorrect.

2. Ln 36-37: Symptoms are not just more common in adults, they also tend to be more severe.

3. Ln 67: The abbreviations NMCSS and SDW should be defined at first instance in the main text, not in the Acknowledgements.

4. Decimals should be depicted with points, not commas.

5. In Table 1 it would be better to present the total number of samples tested in place of the negative ones, again for which marker of HAV infection? Please clarify. At the bottom of the Table, it would be useful to show the total number of samples (from all provinces).

6. The age distribution of HAV cases is shown in Figure 2. What about the gender distribution?

7. Fig. 3: Proportion of HAV cases with respect to what? The (total?) population? The figure itself is of low quality and needs editing.

8. Section “3.3. Correlating Rainfall patterns to HAV cases”: The graphs presented in Fig. 4, especially in the second part are not convincing for a correlation between rainfall and HAV cases. Associations have been reported, for example between heavy rainfall and a higher risk of hepatitis A two weeks after each extra storm day (please see Gullón et al. Association between meteorological factors and hepatitis A in Spain 2010-2014. Environ Int. 2017;102:230-235. doi: 10.1016/j.envint.2017.03.008). Reference number 13 is wrong (compare ln 218 vs. reference mentioned in the list).

9. ln 231-232 (“Some of our data corroborates with Prabdial-Sing (9)): This is a very vague statement. Please elaborate.

Comments on the Quality of English Language

The paper should be proofread carefully. There are some grammatical mistakes (e.g., ln 76, data ... were, not "was").

Author Response

Reviewer 1: Cover letter

Greetings and thank you for providing valuable feedback on what needs to be addressed in the paper before publication.

Most of the revisions that were highlighted were addressed and have been highlighted accordingly.

  1. I have now specified IgM as the marker used in this study in the methods. I unfortunately could not address whether all of our peripheral laboratories were using the exact same test. However, I can confirm that all tests were done for the serological detection of anti-HAV IgM.
  2. The use of HAV & Hepatitis A have been corrected in text
  3. Symptomology in adults’ statement has been amended
  4. NMCSS & SDW have been explained in full at the beginning of the text
  5. Decimals have been adjusted using points in place of commas
  6. Table 1 has been edited such that positive results and negative results have been included
  7. Gender distribution has been added
  8. Figure 3 heading has been adjusted and image quality has been improved
  9. Reference 13 which was incorrectly referenced has been amended
  10. Grammar has been amended.

Appeals

  1. Data from Prabdial-Sings study has been explained in detail following that statement of our data corroborating with her previous study
  2. Regression analysis was just done to observe whether there is a potential association between elevated rainfall vs cases. I adapted this methodology from Guillon et al, as implementation of this methodology hasn’t been recorded in previous HAV papers published in South Africa. Having this data shows that to some degree there is seasonality present however, it does require in depth investigation. Furthermore, this finding highlights the fact that Hepatitis A case fluctuations are indeed a multifactorial event where environmental and climatic factors are implicated amongst many other undiscovered factors.

Reviewer 2 Report

Comments and Suggestions for Authors

This manuscript describes the epidemiology of HAV in South Africa. The manuscript is well written but there are several limitations.

1.       Materials and methods

      HAV markers (IgM or IgG) reported by the surveillance systems must be clearly indicated. Regarding the SDW system (to be defined in the text), this information is lacking. The methods used must be mentionned.

2.       A major limitation is the absence of data on anti-HAV IgG. It could be relevant to determine the IgG seroprevalence in geographic area with the highest incidence.

3.       What are the plausible explanations of geographic differences of cases of HAV ?

Author Response

Reviewer 2: Cover letter

Greetings and thank you for providing valuable feedback on what needs to be addressed in the paper before publication.

All of the revisions that were highlighted were addressed and have been highlighted accordingly

  1. Anti-HAV IgM has been mentioned as the marker used in this study
  2. NMC/SDW has been explained in the methods
  3. We were unable to retrieve IgG data which has been explained in the limitations
  4. Plausible explanations for geographical differences has been mentioned in lines 288-292

Reviewer 3 Report

Comments and Suggestions for Authors

Manuscript ID: viruses-2999602

Title: Exploring the Epidemiological Surveillance of Hepatitis A in South 

Africa: A 2023 Perspective

Authors: Dr. Keveshan Bhagwandin et al.

This was an interesting study in HAV in South Africa. I have two queries in this report. There were no data shown the severity of hepatitis by HAV infection. How about it?

In this area, what age will be recommended for HAV vaccination?

In addition, which is effective on HAV protection, environmental improvement or vaccination?

Minor; 

There were several grammatical mistakes in the text; missing of ‘period’, and an common occurrence. 

Comments on the Quality of English Language

Minor; 

There were several grammatical mistakes in the text; missing of ‘period’, and an common occurrence. 

Author Response

Reviewer 3: Cover letter

Greetings and thank you for providing valuable feedback on what needs to be addressed in the paper before publication.

Most of the revisions that were highlighted were addressed and have been highlighted accordingly.

  1. Based on the WHO position paper, age of vaccination has been mentioned in the conclusions between lines 315-321
  2. Effectiveness between environmental improvements and vaccination has been mentioned between lines 321-322

Appeals

  1. Disease severity could not be determined, and reasons have been mentioned in the limitations which are derived from the way in which NMCSS and SDW was utilized.

Round 2

Reviewer 1 Report

Comments and Suggestions for Authors

Thank you for considering my initial comments. The paper has been improved, but it still needs some work before it can be accepted for publication, in my opinion. Namely:

1.     Table 1: Please check the numbers of the 6th column representing the “Lab positive rate (%)”. For instance, the number in the first row (Eastern Cape) should be 1.77, not 1.80 and in the second row (Free State) it should be 2.28, not 2.33.

2.     Figure 1. Does this figure show the incidence of IgM anti-HAV (per 100,000 population?) as stated in the legend, or anti-HAV cases as indicated in the legend of the y-axis? Please correct accordingly.

3.     Figures 2 and 3 essentially show the same results and should be combined into one figure.

4.     Figure 4: Were any of the “noticeable differences” (observed in Gauteng, Kwa-Zulu Natal and Limpopo provinces, where HAV was detected more often in males) statistically significant? A proper analysis should be conducted.

5.     Ln 184 “3.3. Correlating Rainfall patterns to Hepatitis A cases”: The evidence is very weak. Please replace “correlating” with “Linking”. A description of the undertaken analysis should be added to the Methods section.

6.     Legend of Figure 5. (Illustration and correlation of monthly rainfall compared to HAV cases in the 3 most burdened provinces): Please replace the word “correlation” with “relation”).

Minor

1. Ln 31: Please replace “HAV” with “Hepatitis A virus (HAV)”.

2. “Hepatitis A” within the text (ln 45, 56, 81, 184, 317) the first letter of the word “hepatitis” should not be capitalized (“hepatitis A”).

Comments on the Quality of English Language

ln 82 (Methods): Data were analyzed...

Author Response

Thank you for the noting revisions needed 

All revisions have been addressed and are highlighted in green 

Reviewer 2 Report

Comments and Suggestions for Authors

The answers are satisfactory. The manuscript has been improved.

Author Response

All changes have been highlighted in green 
